# CTTN Overexpression Confers Cancer Stem Cell-like Properties and Trastuzumab Resistance via DKK-1/WNT Signaling in HER2 Positive Breast Cancer

**DOI:** 10.3390/cancers15041168

**Published:** 2023-02-11

**Authors:** So-Jeong Moon, Hyung-Jun Choi, Young-Hyeon Kye, Ga-Young Jeong, Hyung-Yong Kim, Jae-Kyung Myung, Gu Kong

**Affiliations:** 1Department of HY-KIST Bio-Convergence, Hanyang University, Seoul 04763, Republic of Korea; 2Department of Pathology, College of Medicine, Hanyang University, Seoul 04763, Republic of Korea

**Keywords:** HER2 positive breast cancer, cancer stem cells, CTTN, trastuzumab resistance

## Abstract

**Simple Summary:**

Cortactin (CTTN) is an actin-binding protein that is mainly known for its ability to promote cancer progression. It is unclear, however, whether CTTN affects tumor initiation and anti-tumor drug resistance in breast cancer. Here, we investigated the potential role of CTTN as a novel poor prognostic biomarker and possible therapeutic target of trastuzumab resistance for HER2 positive breast cancer. Our findings showed that CTTN enhances tumor initiation and increases anti-HER2 drug resistance by activating the DKK-1/Wnt/β-catenin signaling pathway. CTTN-induced cancer stem cell-like properties were reversed by treatment of β-catenin/TCF inhibitor. Furthermore, combinational treatment with trastuzumab and β-catenin/TCF inhibitor overcame CTTN-induced trastuzumab resistance. These results suggest that combined treatment of trastuzumab and Wnt signaling inhibitors could be an effective therapeutic strategy to treat HER2+ breast tumors expressing high levels of CTTN. Collectively, we suggest that CTTN could be a potential target for treatment of trastuzumab resistance in HER2 positive breast cancer patients.

**Abstract:**

Background: Despite the therapeutic success of trastuzumab, HER2 positive (HER2+) breast cancer patients continue to face significant difficulties due to innate or acquired drug resistance. In this study we explored the potential role of CTTN in inducing trastuzumab resistance of HER2+ breast cancers. Methods: Genetic changes of CTTN and survival of HER2+ breast cancer patients were analyzed in multiple breast cancer patient cohorts (METABRIC, TCGA, Kaplan-Meier (KM) plotter, and Hanyang University cohort). The effect of CTTN on cancer stem cell activity was assessed using the tumorsphere formation, ALDEFLUOR assay, and by in vivo xenograft experiments. CTTN-induced trastuzumab resistance was assessed by the sulforhodamine B (SRB) assay, colony formation assays, and in vivo xenograft model. RNA-seq analysis was used to clarify the mechanism of trastuzumab resistance conferred by CTTN. Results: Survival analysis indicated that CTTN overexpression is related to a poor prognosis in HER2+ breast cancers (OS, *p* = 0.05 in the Hanyang University cohort; OS, *p* = 0.0014 in KM plotter; OS, *p* = 0.008 and DFS, *p* = 0.010 in METABRIC). CTTN overexpression-induced cancer stem cell-like characteristics in experiments of tumorsphere formation, ALDEFLUOR assays, and in vivo limiting dilution assays. CTTN overexpression resulted in trastuzumab resistance in SRB, colony formation assays, and in vivo xenograft models. Mechanistically, the mRNA and protein levels of DKK-1, a Wnt antagonist, were downregulated by CTTN. Treatment of the β-catenin/TCF inhibitor reversed CTTN-induced cancer stem cell-like properties in vitro. Combination treatment with trastuzumab and β-catenin/TCF inhibitor overcame trastuzumab resistance conferred by CTTN overexpression in in vitro colony formation assays. Conclusions: CTTN activates DKK-1/Wnt/β-catenin signaling to induce trastuzumab resistance. We propose that CTTN is a novel biomarker indicating a poor prognosis and a possible therapeutic target for overcoming trastuzumab resistance.

## 1. Introduction

Breast cancer is the most common cancer in women worldwide. It can be classified into three major subtypes based on molecular and histological findings: estrogen receptor positive (ER+), human epidermal growth factor receptor 2 positive (HER2+), and triple negative. HER2+ breast cancer, which represents approximately 15–20% of all breast cancers, is known to be associated with poor prognosis and an aggressive phenotype [1,2]. HER2+ breast cancer is generally treated with trastuzumab. Even though this treatment has improved the survival of HER2+ breast cancer patients, its clinical efficacy is limited by drug resistance [3,4].

Chr11q13 amplification occurs in about 15% of primary breast cancers and is common in human cancers. Many genes are located in the chr11q13 amplicon, including FGF3, FGF4, CCND1 and CTTN [5,6]. Cortactin (CTTN) is an actin binding protein which activates actin polymerization by binding simultaneously to the Arp2/3 complex and F-actin and stabilizing filament branches produced by the Arp2/3 complex. CTTN is a candidate for driving the 11q13 amplicon and is widely known to enhance invasion and metastasis, resulting in poor clinical outcomes. Its overexpression is frequently observed in many cancers including breast cancer [6,7]. Phosphorylation of CTTN promotes migration, and CTTN mediates CD44-promoted invasion by bone marrow endothelial cells [8,9,10]. It has also been shown to contribute to colorectal cancer and melanoma and to drug resistance in head and neck squamous cell carcinoma [11,12,13]. Despite its potential as an oncogene, its roles in breast cancer initiation and in resistance to cancer drugs including trastuzumab remain largely unknown.

Numerous mechanisms of trastuzumab resistance have been suggested, including PTEN loss, activation and inactivation of alternative cancer signaling pathways, and altered expression of ligands of the EGFR family and of the tyrosine kinase receptor [14]. In addition, accumulating evidence suggests that an increased number of cancer stem cells (CSCs) in HER2+ breast cancer is linked to trastuzumab resistance [15]. CSCs are a subset of the cancer cell population that grow continuously; they are resistant to cancer treatment, are aggressive in terms of invasion and metastasis, and have self-renewal ability [16,17]. Several emerging therapeutic approaches to overcoming trastuzumab resistance in HER2+ breast cancer involve reducing or eliminating CSCs [18,19,20,21,22,23]. Since targeting breast CSCs is thought to be a promising therapeutic strategy for HER2+breast cancer treatment, it is important to identify novel ways of overcoming the trastuzumab resistance of CSCs.

## 2. Materials and Method

### 2.1. Patients

In total, 59 formalin-fixed and paraffin-embedded (FFPE) HER2+ breast cancer tissues were obtained from patients who received curative surgery at Hanyang University Hospital, Seoul, Korea, from 2003 to 2016. HER2 status and ER status were evaluated according to ASCO/CAP guidelines, as previously described [24]. Clinicopathological data such as patient age, histologic grade, tumor size, and evidence of lymphatic metastasis were obtained by thorough review of the medical and pathological records. This study was approved by the Institutional Review Board of Hanyang University Hospital (IRB No. HYUH 2021-12-014-001).

### 2.2. Immunohistochemistry and Interpretation

Immunohistochemistry (IHC) was carried out on TMA (Tissue micro array) 3 mm core sections using a Benchmark XT automated staining system (Ventana Medical Systems, Tucson, AZ, USA). Primary anti-CTTN antibody (11381-1-AP, 1:400, Proteintech) was used according to the manufacturer’s instructions. IHC staining was interpreted by two pathologists. Immunoreactive scores (IRS) were obtained by multiplying intensity scores by percentages of immunoreactive cells and were used to assess semi-quantitatively the cytoplasmic staining of the tumor cells. The intensity of staining was categorized as 1 to 3 (1: weak, 2: moderate, and 3: strong). Percentages of immunoreactive cells were divided into 4 levels: 0 (0%), 1 (1–25%), 2 (26–50%), 3 (51–75%), and 4 (>75%). IRS ranged between 4 and 12. We divided IRS into two categories: weak expression (IRS ≤ 8) and strong expression (IRS > 8) and used them in all statistical analyses.

### 2.3. Cell Culture and Reagents

MCF10A, ZR75-30, MDA-MB-453, BT20, JIMT-1, MDA-MB-231, and HEK293T cells were obtained from the American Type Culture Collection (ATCC, Manassas, VA, USA). MCF7, T47D, ZR75-1, SKBR3, BT474, HCC-1954, HCC1419, and HS578T cells were purchased from the Korean Cell Line Bank. MCF7, T47D, MDA-MB-231, and HEK293T cells were cultured in DMEM (Welgene, Daegu, Republic of Korea) supplemented with 10% FBS, and SKBR3, BT474, HCC-1954, HCC-1419, JIMT-1, and HS5788T cells were cultured in RPMI (Welgene) supplemented with 10% FBS at 37 °C in a 5% CO2 atmosphere. Trastuzumab was obtained from Roche (Basel, Switzerland). The β-Catenin/TCF inhibitor FH-535 was purchased from Sigma-Aldrich (St. Louis, MO, USA).

### 2.4. Transfection of siRNAs and cDNA

siRNAs against CTTN (#1, 50-GAAUAUCAGUCGAAACUUUtt-30, cat no. 2017-2; #2, 50-CGACAAGGACAAAGUGGAUtt-30, cat no. 2017-3) were purchased from Bioneer (Daejeon, Republic of Korea) and transfected into breast cancer cells with Lipofectamine 2000 (Invitrogen, Carlsbad, CA, USA) over 48 h.

### 2.5. Lentiviral Infection and Generation of Stable Cell

Lentiviral pLVX-puro vector was purchased from Clontech (Mountain View, CA, USA) and used for cloning human CTTN cDNA. CTTN shRNAs (cat no. RHS4531-EG2017; Clone IDs, shRNA #1: V3LHS_308287 and shRNA #2: V2LHS_43343) were inserted into pGIPZ vector. Lentiviral particles carrying CTTN cDNA or shRNAs were produced as previously described [25]. To establish CTTN-overexpressing and CTTN-knockdown cell lines, respectively, 6 μg/mL polybrene (Sigma-Aldrich) was used for infections with lentiviruses carrying CTTN cDNA or shRNAs. An amount of 2 μg/mL puromycin (Sigma-Aldrich) was used to select infected cells.

### 2.6. Immunoblotting

Radioimmunoprecipitation (RIPA) buffer with phosphatase-inhibitor cocktails and protease-inhibitor was used to obtain cell lysates. Immunoblotting was carried out as previously described [26]. The following antibodies were used: β-catenin (F5682) from BD Biosciences (Franklin Lakes, NJ, USA); CyclinD1 (SC-8396) and DKK-1 (SC-25516) from Santa Cruz Biotechnology (Dallas, TX, USA); GSK-3β (9315S) and p-GSK-3β (9336) from Cell Signaling Technology (Cell Signaling Technology, Beverly, MA, USA); β-actin (MAB1501R) from Millipore (Billerica, MA, USA); α-tubulin (GTX628802) and SP1 (GTX110593) from GeneTex (Irvine, CA, USA).

### 2.7. The Sulforhodamine B Colorimetric (SRB) Assay

The assay kit obtained from Sigma-Aldrich was used. SKBR3 (3 × 10^3^ cells/well), BT474 (5 × 10^3^ cells/well), JIMT-1 (5 × 10^3^ cells/well) and HCC-1954 (3 × 10^3^ cells/well) cells in RPMI (Welgene) containing 10% FBS were plated in a 96-well plate and incubated with trastuzumab for five days. After fixing in trichloroacetic acid (TCA) for an hour at 4 °C, cells were stained for 30 min with 0.4% sulforhodamine B, destained in 1% acetic acid, dissolved in 10 mM Tris, and examined at 490 nm.

### 2.8. Colony Formation Assay

SKBR3 (3 × 10^3^ cells/well) and BT474 (2 × 10^4^ cells/well) cells were cultured in 24-well plates, and the indicated drugs were added the following day. Drugs and media were changed every 3 days, and incubation was continued until the cells in control plates were confluent. The plates were PBS-washed and fixed with methanol for 5 min. Following another PBS rinse, colonies were stained with 0.04% crystal violet at room temperature for 30 min and rinsed with sterile distilled water.

### 2.9. Quantitative Real-Time Polymerase Chain Reaction (qRT-PCR)

Total RNA was extracted and RT–PCR was carried out as described [27]. mRNA levels of target genes were measured using SYBR Green master mix (Applied Biosystems, Foster City, CA, USA) and a CFX Connect Real-Time PCR Detection System (Bio-Rad, Hercules, CA, USA) and normalized with the mRNA levels of GAPDH. The following primers were used in qRT-PCR: WNT6, 5′-GGTTATGGACCCTACCAGCA-3′ and 5′-GGTTATGGACCCTACCAGCA-3′; DKK-1, 5′-GATCATAGCACCTTGGATGGG-3′ and 5′-GGCACAGTCTGATGACCG-3′; RAC2, 5′-ACAAGGACACCATCGAGAAACT-3′ and 5′-CACTCCAGGTATTTCACCGAG-3′; RAC3, 5′-TCCCCACCGTTTTTGACAACT-3′ and 5′-GCACGAACATTCTCGAAGGAG-3′; FGF22, 5′- CTCCACTCACTTCTTCCTGCG-3′ and 5′-CTGCTTTGATGACCACGACG-3′; CCNA1, 5′- GCTGCTAACTGCAAATGGGC-3′ and 5′-GCTGGAGGGAAGGCATTTTC-3′; CD36, 5′-GCAAAA CGGCTGCAGGTCAA-3′ and 5′-TTCTCATCACCAATGGTCCCAG-3′; ELOVL4, 5′-GTAGTGTCCACGGCACTCAA-3′ and 5′-ATTTTGGACCCAGCCACACA-3′; PPARG, 5′-TGCAGTGGGGATGTCTCATA-3′ and 5′-TGGTCAGCGGGAAGGACTTTA-3′; GAPDH, 5′-CATGTTCCAATATGATTCCA–5‘ and 5′-CCTGGAAGATGGTGATG-3′.

### 2.10. Flow Cytometry

To assess numbers of cells with high ALDH activity, which is a trait of CSC-like cells, ALDEFLUOR assays were carried out [28]. Briefly, 1 mL of ALDEFLUOR assay buffer containing the ALDH substrate, BODIPY aminoacetaldehyde (BAAA), was added to 1 × 10^6^ cells and incubated at 37 °C for 20 min. Experimental groups of cells were compared to negative control cells treated with the specific ALDH inhibitor, diethylaminobenzaldehyde (DEAB). Viable cells and negative controls served as the basis for the sorting gates, and an FACS Diva (BD Biosciences) was used to measure the population with strong ALDH activity.

### 2.11. Tumorsphere Analysis

To measure tumorsphere formation [26] BT474 (1 × 10^4^ cells/well), SKBR3 (5 × 10^3^ cells/well), and HCC-1954 (5 × 10^3^ cells/well) cells were plated in DMEM-Gluta MAX medium (Invitrogen) containing 2% B27 (Invitrogen), 4 ng/mL heparin (Sigma-Aldrich), 20 ng/mL epithelial growth factor (Sigma-Aldrich), and 20 ng/mL basic fibroblast growth factor (PeproTech, Rocky Hill, NJ, USA) in 6-well ultra-low attachment surface plates (Corning, Corning, NY, USA). JIMT-1 (1 × 10^4^ cells/well) cells were similarly seeded in DMEM-Gluta MAX medium supplemented with 2% B27, 4 ng/mL heparin, 20 ng/mL EGF, 20 ng/mL bFGF, 5 mg/mL insulin, and 0.5 mg/mL hydrocortisone (Sigma-Aldrich), in ultra-low attachment surface plates. An inverted microscope was used to assess and quantify tumorsphere formation after 5 days.

### 2.12. Orthotopic Xenografts

Five-week-old female NOD/SCID mice were obtained from KOATECH (Pyeongtaek, Republic of Korea) and Korea Research Institute of Bioscience and Biotechnology (Deajeon, Republic of Korea). Mice (*n* = 7 per group) were orthotopically injected with serially diluted SKBR3 and HCC-1954 cells in limiting dilution assays. L-Calc software (http://www.stemcell.com) was used to calculate numbers of tumor-initiating cells (TIC). To evaluate the effects of trastuzumab, control and CTTN-overexpressing SKBR3 cells (2 × 10^6^ cells) suspended in Matrigel were injected into the fat pads of mice. The mice were intraperitoneally injected with trastuzumab (20 mg/kg) or phosphate-buffered saline as vehicle twice per week after the tumors reached 100 mm^3^. Tumor size was measured twice weekly with calipers, and tumor volume was estimated as 1/2 × (long diameter) × (short diameter)^2^.

### 2.13. RNA Sequencing (RNA-Seq) Analysis

RNA-seq analysis of control and CTTN-overexpressing SKBR3 cells was performed by Macrogen (Seoul, Republic of Korea) as previously described [29]. Briefly, total RNA was extracted with Trizol (Life Technologies, Carlsbad, CA, USA), and a bioanalyzer supplied with the Agilent RNA 6000 Pico Kit (Agilent Technologies, Santa Clara, CA, USA) was used to examine RNA integrity. The manufacturer’s instructions were followed while processing the extracted total RNA. A TruSeq standarded mRNA sample preparation kit (Illumina, San Diego, CA, USA) was used to create an mRNA sequencing library, and a KAPA Library Quantification kit and NovaSeq6000 sequencer (Illumina) were used to sequence all samples. After being transformed into sequence data by base calling, the raw data were stored in FASTQ format. For PCR and sequencing adapters, the paired end reads of the 6 distinct samples were trimmed using Cutadapt (http://cutadapt.readthedocs.io/en/stable/; version1.16 (accessed on 22 January 2022)). STAR (version 2.6.0c) was used to align trimmed reads to the hg19 human reference, and Cufflinks was used to generate gene-level read counts. EdgeR (version 3.22.3) was employed to identify significant differential expression [30] involving at least 1.5-fold changes between the control and CTTN-overexpressing groups. Differentially expressed genes were categorized using version 7.5.1 of Gene sets (http://software.broadinstitute.org/gsea/msigdb/index.jsp (accessed on 4 may 2022.)) of the Molecular Signatures Database using version 4.2.3 of GSEA. R language (https://www.r-project.org/; version 3.5.1 (accessed on 22 January 2022.)) was used to generate the heatmap of differential gene expression.

### 2.14. Public Data

The Cancer Genome Atlas (TCGA) and METABRIC, which are publicly accessible datasets concerning patients with breast cancer, were acquired from cBioPortal (http://www.cbioportal.org/) and reanalyzed. The Cancer Cell Line Encyclopedia dataset and OncoPrints showing genetic alteration of CTTN in the cohorts of TCGA and METABRIC were produced using cBioPortal. Disease-free survival (DFS) and overall survival (OS) of HER2+ breast cancer patients in the METABRIC dataset were analyzed with Cancer Target Gene Screening (CTGS; http://ctgs.biohackers.net) [31]. The cut-off point with the highest significance for each gene was selected using the best *p*-value obtained with CTGS in order to classify patient groups based on the expression levels of genes at 11q13 in METABRIC. Then, using the R programming package, the hazard ratios (HR) and 95% intervals (Cis) for the top 10 genes in a Cox proportional regression model were displayed in forest plots. Autoselect best cut-off of CTTN (201059_at) was used in KM plotter. The public datasets of expression-profiling of HER2+ breast cancer patients treated with adjuvant trastuzumab (GSE58984 [32], GSE5534 [33], GSE50948 [34] were obtained from Gene Expression Omnibus (GEO) and reanalyzed. R language was used to generate a heatmap of the differentially expressed genes.

### 2.15. Statistical Analysis

The unpaired Student’s *t*-test was used to determine whether differences between pairs of groups were statistically significant. One-way ANOVA followed by a post hoc LSD test (equal variance) and repeated-measure (RM) ANOVA were used to assess multiple group comparisons and repeated measures, respectively. Kaplan–Meier plots for survival analysis were created using the Kaplan–Meier method with log-rank tests. Pearson’s correlation coefficient was used to determine r values. All *p*-values were two sided. *p* < 0.05 was considered statistically significant.

## 3. Results

### 3.1. Genetic Alteration of CTTN Is Associated with Poor Prognosis of HER2 Positive Breast Cancer Patients

We analyzed and sorted genes in 11q13 as reported in our previous study [29,31]. The expression levels of genes located in 11q13 and their genetic alterations were analyzed using the METABRIC (*n* = 247) and TCGA (*n* = 120) breast cancer datasets. About 20% of HER2+ breast cancers had abnormal CTTN copy numbers and/or expression, and the frequencies for CCND1 were similar, while alterations in the other genes were found at relatively low frequencies (Figure 1A). Genes linked to poor survival outcomes in HER2+ breast cancer were sorted among those genes exhibiting positive correlations between expression levels and copy number alterations (r ≥ 0.3). Notably, CTTN had the highest hazard ratio for both overall survival (OS) and disease-free survival (DFS) (HR, 1.994; *p* = 0.009, HR, 2.005; *p* = 0.018, respectively, Figure 1B). CTTN expression was significantly higher in ER+/HER2+ breast cancer (Figure 1C, right), and high expression levels were significantly correlated with amplification of CTTN (*r =* 0.75, *p <* 0.001; Figure 1C, left) and HER2 expression (*r* = 0.2, *p* < 0.001; Figure 1D). CTTN overexpression was associated with poorer survival outcomes in HER2+ breast cancer patients in METABRIC (OS, *p* = 0.008; DFS, *p* = 0.010; Figure 1E, top), our cohorts (OS, *p* = 0.05; Figure 1F), and Kaplan–Meier plotter (OS, *p* = 0.0014), and it showed a tendency toward shorter relapse-free survival (RFS) in Kaplan–Meier plotter (*p* = 0.066; Figure 1E, bottom). Taken together, these data suggest that amplification and overexpression of CTTN are biomarkers for a poor prognosis in HER2+ breast cancer.

### 3.2. CTTN Induces Cancer Stem Cell Properties and Resistance to Trastuzumab in HER2+ Breast Cancer

We established lentiviral-induced CTTN overexpression and shRNA-mediated CTTN knockdown in various breast cancer cells to clarify the functional significance of CTTN in HER2+ breast cancer (Appendix A). Because CTTN overexpression is associated with poor disease-free survival of HER2+ breast cancer patients, we assessed the impact of CTTN on CSC activity in HER2+ breast cancer. Fluorescence-activated single cell sorting (FACS) was used to detect cells with high aldehyde dehydrogenase (ALDH). The results showed that overexpression of CTTN increased numbers of self-renewing CSCs in trastuzumab-sensitive SKBR3 and BT474 cells, while knockdown of CTTN in trastuzumab-resistant JIMT-1 and HCC-1954 cells resulted in a reduction of CSC-like populations (Figure 2A). Likewise, primary tumorsphere formation was enhanced in CTTN-overexpressing cells, while CTTN knockdown reduced tumorsphere formation (Figure 2B). Furthermore, tumors developed rapidly in mice orthotopically xenografted with CTTN-overexpressing SKBR3 cells, while they were reduced in mice bearing CTTN-knockdown HCC-1954 cells (Table 1). We also examined tumor-free rates in xenografted mice and noted a significant reduction in tumor-free rates in mice harboring CTTN-overexpressing SKBR3 cells (*p* = 0.048; Figure 2C, left). Conversely, tumor-free rates were significantly elevated in mice harboring CTTN-knockdown HCC-1954 cells (*p* = 0.001; Figure 2C, right). We also investigated whether CTTN had an impact on the response of breast cancer cells to anti-HER2 therapy. 

CTTN Overexpression resulted in insensitivity to trastuzumab in trastuzumab-sensitive HER2+ breast cancer cell lines. However, CTTN knockdown did not result in increased trastuzumab sensitivity in cells that were trastuzumab-resistant (Figure 2D). When CTTN-overexpressing SKBR3 cells and control SKBR3 cells were assayed for colony formation (Figure 2E) or received orthotopic xenografts (Figure 2F), both displayed trastuzumab resistance. Collectively, these findings demonstrate that CTTN enhances CSC activity and induces trastuzumab resistance, suggesting that CTTN is a possible oncogenic driver and a potential target for overcoming trastuzumab resistance in HER2+ breast cancer.

Control (CON) or CTTN-overexpressing (CTTN) SKBR3 cells and control (shCON) or CTTN-knockdown (shCTTN) HCC-1954 cells were injected into the mammary fat pads of NOD/SCID mice at doses ranging from 1000 to 100,000 cells. L-Calc software (Stemcell Tech, http://www.stemcell.com) was used to calculate frequencies of tumor-initiating cells (TIC).

### 3.3. CTTN Downregulates DKK-1 Expression and Activates the Wnt Signaling Pathway

To further understand how CTTN affects CSC activity and trastuzumab resistance in HER2+ breast cancer, we performed an RNA-seq analysis of CTTN-overexpressing SKBR3 cells. CTTN overexpression was associated with overexpression of gene sets related to stem cells, breast cancer relapse, EMT, and the b-catenin/TCF complex (Figure 3A,B), and these effects were validated by quantitative real-time polymerase chain reaction (qRT-PCR) (Figure 3C). We also identified CTTN-target genes associated with trastuzumab resistance by comparing this RNA-seq data with 3 GEO datasets (GSE58984, GSE55348, GSE50948) of trastuzumab-treated patients (Figure 3D, Appendix A). Prompted by the GSEA results, we found that the Wnt signaling antagonist DKK-1 was downregulated by CTTN overexpression. Wnt/β-catenin signaling has been identified as a key mechanism regulating CSCs [35]. Therefore, we measured the protein and mRNA levels of DKK-1 in CTTN-overexpressing and -knockdown cell lines by western blotting (Figure 3E and Appendix A) and real time qPCR (Figure 3F), respectively. To confirm these findings, we then investigated whether CTTN activated the Wnt signaling pathway and found that phosphorylated GSK3-β and β-catenin were elevated in CTTN-overexpressing SKBR3 cells and decreased in CTTN-knockdown JIMT-1 cells (Figure 3E). Furthermore, overexpression of CTTN stimulated translocation of β-catenin to the nucleus, while CTTN knockdown resulted in reduced nuclear localization of b-catenin (Figure 3G). These findings collectively imply that CTTN downregulates the expression of DKK-1 thus activating the Wnt/β-catenin/TCF signaling pathway.

### 3.4. Inhibition of Wnt Signaling Reverses CTTN-Induced CSC-like Properties and Trastuzumab Resistance

We next explored whether the Wnt signaling pathway was responsible for the increased breast CSC-like cell numbers and trastuzumab resistance caused by CTTN overexpression. Increased levels of c-Myc and CyclinD1 in CTTN overexpressing SKBR3 and BT474 cells were reversed by treatment with the β-catenin/TCF inhibitor, FH535 (Figure 4A). Furthermore, the increase in the cell population with high ALDH activity due to CTTN-overexpression was also abolished by FH535 treatment (Figure 4B). Similarly, treatment with FH535 resulted in reduced tumorsphere formation by CTTN-overexpressed SKBR3 and BT474 cells (Figure 4C). Since it has been suggested that CSCs are responsible for the resistance to anti-HER2 drugs [15], we tested whether the CTTN-induced Wnt signaling affected the response to trastuzumab. In colony formation assays of CTTN-overexpressing SKBR3 and BT474 cells, combination treatment with trastuzumab and FH535 overcame trastuzumab resistance (Figure 4D).

Thus, we have shown that CTTN induces CSC activity, tumor growth, and trastuzumab resistance in HER2+ breast cancer. Mechanistically, CTTN appears to downregulate DKK-1 and thereby activate the Wnt signaling pathway, which then induces CSC-like behavior and trastuzumab resistance (Figure 4E).

## 4. Discussion

Our findings clarify the role of CTTN in inducing tumor initiation and trastuzumab resistance in HER2+ breast cancer. CTTN overexpression is a marker of a poor prognosis in HER2+ breast cancer. Mechanistically, CTTN downregulates DKK-1, a Wnt antagonist, in HER2+ breast cancer, resulting in activation of the Wnt signaling pathway and expansion of the number of CSCs, as well as resistance to trastuzumab (Figure 4E). Treatment with the Wnt signaling inhibitor was able to overcome CTTN-induced trastuzumab resistance in HER2+ breast cancer. This evidence suggests that CTTN could be clinically useful for overcoming the trastuzumab-resistance of HER2+ breast cancers.

Various groups have previously suggested that innate and acquired resistance to trastuzumab is related to PI3K signaling, IGF1R and EGFR, and expression of p95HER2 [3,14]. In addition, emerging evidence suggests that it is closely associated with the presence of breast CSCs (BCSCs). Deregulation of several pathways involved in regulating BCSCs, including the Notch, TGF-β/SMAD, and Wnt/β-catenin pathways, can result in an increase or transition of BCSCs. Targeting these pathways in combination with trastuzumab treatment was able to overcome trastuzumab resistance [15]. Consistent with this, our data suggest that CTTN enhances trastuzumab resistance by downregulating DKK-1, thereby activating the Wnt signaling pathway. It has been reported that overexpression of WNT3 activates the Wnt/β-catenin signaling pathway, resulting in transactivation of EGFR and provoking an EMT-like phenotype, which in turn could underlie the trastuzumab resistance of HER2+ breast cancers [36]. In gastric cancer, the canonical Wnt/β-catenin signaling pathway was found to be activated in trastuzumab-resistant cancer cells [37]. In agreement with these results, we demonstrated that treatment with the Wnt signaling inhibitor, FH535, overcame the trastuzumab resistance of CTTN-overexpressing HER2+ breast cancer cells. Thus, our results suggest that trastuzumab and a TCF/β-catenin inhibitor could be used in combination to treat HER2+ breast tumors expressing high levels of CTTN.

Amplification of chromosome locus 11q13 has been related to a poor prognosis in many human cancers, including breast cancer [38,39,40]. CTTN, which is in this region, has been identified as a putative promoter of tumorigenesis and is mainly involved in processes underlying tumor progression, such as cell migration, invasion, and tumor metastasis [8,9,10,11,12,13,41,42]. However, little is known about its ability to initiate tumors and induce anti-cancer drug resistance. Wei et al. have reported that in melanoma low RNF128 levels promote EMT and stemness via Wnt signaling by protecting CD44 and CTTN from degradation [12]. In agreement with this, we report that CTTN induces CSC-like properties via Wnt signaling in breast cancer, suggesting that CTTN is involved in tumor initiation as well as progression. There are several reports that overexpression of CTTN can induce resistance to anti-cancer drugs. In head and neck squamous cell carcinoma (HNSCC), CTTN induces gefitinib resistance by attenuating EGFR and c-MET degradation [13]. Another study has revealed that the β1 integrin/FAK/cortactin/JNK1 pathway is essential for radiation resistance in HNSCC cells [43]. Our data suggest that CTTN plays a crucial role in resistance to anti-cancer treatments such as chemotherapy and radiotherapy. To the best of our knowledge, we are the first to propose a role of CTTN in the trastuzumab resistance of HER+ breast cancer.

## 5. Conclusions

In this study, we have revealed that CTTN not only promotes cancer progression but also plays a crucial role in tumor initiation and anti-cancer drug resistance in HER2+ breast cancer. CTTN enhanced the number of CSCs and conferred trastuzumab resistance by downregulating DKK-1 and activating the Wnt signaling pathway. We propose that HER2+ breast cancer patients with high CTTN expression could benefit from combination therapy with trastuzumab and Wnt signaling inhibitors such as FH535. In conclusion, CTTN is a biomarker of a poor prognosis and a possible target for treating the trastuzumab resistance of HER2+ breast cancers.

## Figures and Tables

**Figure 1 cancers-15-01168-f001:**
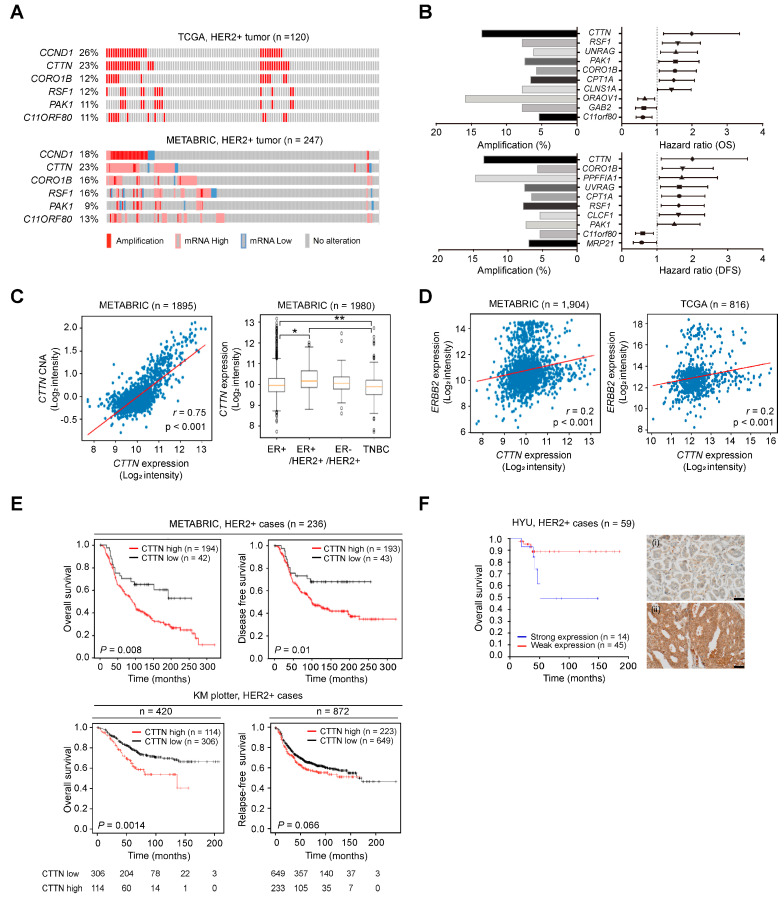
Genetic alterations of CTTN and their clinical impact in HER2 positive breast cancer. (**A**) Oncoprints representing the gene amplification and expression levels of genes at 11q13 in the indicated breast cancer datasets. (**B**) Forest plots representing the hazard ratios (HRs) of genes located at 11q13 according to the overall survival (OS) (**top right**) and disease-free survival (DFS) (**bottom right**) of HER2+ breast cancer patients from the METABRIC dataset. Genes in 11q13 were nominated based on hazard ratios (CTTN, HR (OS), 1.994; *p* = 0.009, HR (DFS), 2.005; *p* = 0.018). Bar graphs represent the percentage amplifications of the indicated genes (**left**). (**C**) Scatter plot displaying the correlation between copy number alteration (CNA) and expression of CTTN in METABRIC (**left**). Pearson’s correlation coefficient was used to determine r values. The box plots display CTTN mRNA levels in selected subtypes from METABRIC (**right**). The vertical bars from the boxes (whiskers) indicate lowest and highest values. The horizontal lines (red) inside the boxes indicate medians. The 25th and 75th percentiles are represented by the bottom and top of the boxes, respectively. *p*-values were determined by one-way ANOVA followed by post hoc Tukey tests. * *p* < 0.05, ** *p* < 0.01 (**D**) Scatterplots displaying the correlation between expression of CTTN and HER2. Pearson’s correlation coefficient was used to determine the r value. *p*-values were determined by one-way ANOVA with post hoc LSD tests. (**E**) Using the Kaplan–Meier method and the log-rank test, the survival of HER2+ breast cancer patients was analyzed based on CTTN expression in METABRIC (**top**) and KM plotter (**bottom**). (**F**) Analysis of the OS of HER2+ breast cancer patients in the Hanyang cohort (HYU, Hanyang University) according to CTTN expression. Groups were stratified by IHC-based CTTN protein expression. *p*-values were determined by the Kaplan–Meier method with the log-rank test. Representative photomicrographs of intensity scores for CTTN staining in HER2 positive breast cancer (right). ((**i**): weak, (**ii**): strong, original magnification ×400, scale bars, 50 µm).

**Figure 2 cancers-15-01168-f002:**
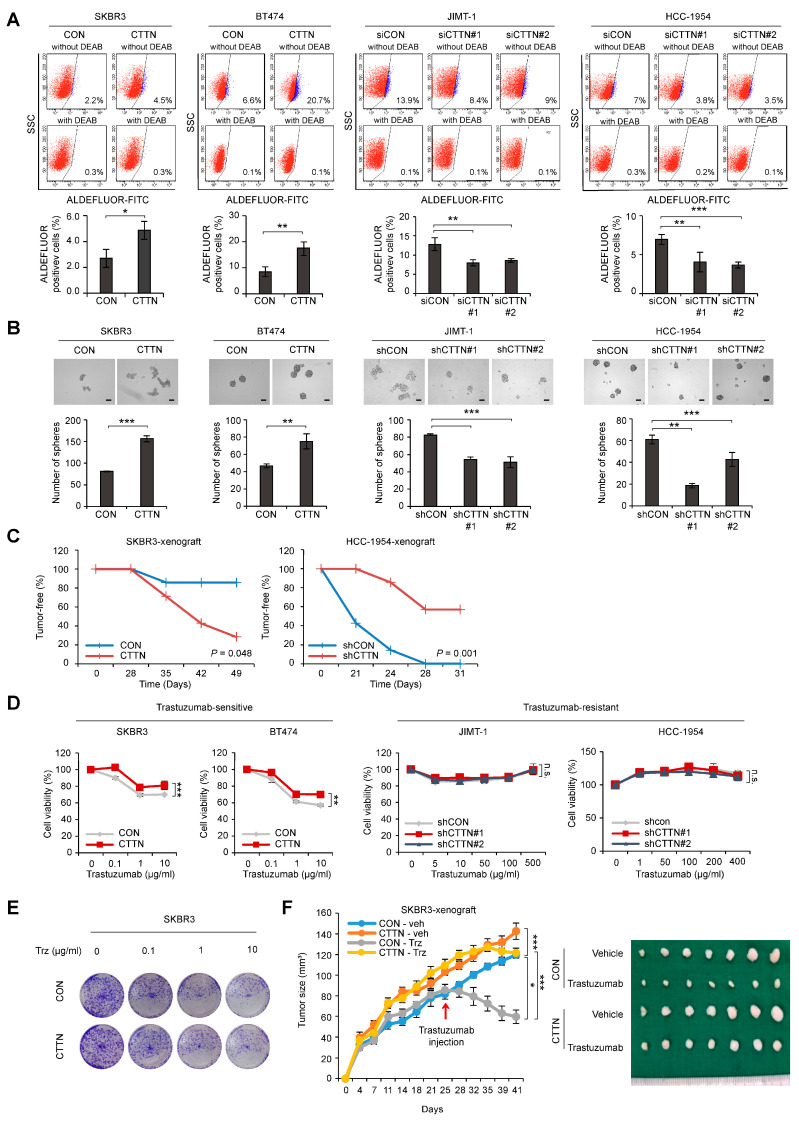
Effect of CTTN overexpression on breast CSC activity and response to trastuzumab in HER2 positive breast cancer. (**A**) The ALDEFLUOR assay was used on the indicated cell lines to analyze cell populations showing high ALDH activity. Cells were either treated with the ALDH inhibitor DEAB (negative control) or left untreated (experimental groups) and then incubated with the ALDEFLUOR substrate (BAAA). The percentage of the population that was ALDEFLUOR-positive was analyzed by FACS from the shift of fluorescent cells observed in the dot plots of samples not exposed to DEAB. Data are means ± SD of three independent samples. *p*-values were determined using two-tailed Student’s *t*-tests for SKBR3 and BT474 cells or one-way ANOVA followed by a post hoc LSD test for JIMT-1 and HCC-1954 cells. * *p* < 0.05, ** *p* < 0.01, *** *p* < 0.001 versus controls (CON or siCON). CON, control; CTTN, CTTN overexpression; siCON, siRNA control; siCTTN, CTTN siRNA; with DEAB, DEAB-treated samples; without DEAB, not exposed to DEAB. (**B**) Tumorsphere formation assays were used to examine the self-renewal ability of CSCs in the indicated cell lines. After 5 days, the number of tumorspheres (>100 μm in diameter) was counted. Data are presented as the means ± SD of three biological replicates. *p*-values were determined by two-tailed Student’s *t*-tests (BT474 cells) or one-way ANOVA with a post hoc LSD test (HCC-1954 cells). ** *p* < 0.01, *** *p* < 0.001 versus controls (CON or shCON). CON, empty vector; CTTN, CTTN overexpression; shCON, shRNA control; shCTTN, CTTN knockdown. Scale bar 100 μm. (**C**) Tumor-free survival curves of mice orthotopically injected with the indicated cells. *p*-values were calculated using the Kaplan–Meier method with the log-rank test. CON, empty vector; CTTN, CTTN overexpression; shCON, shRNA control; shCTTN, CTTN knockdown. (**D**) Sulforhodamine B (SRB) assays were used to determine the viability of the indicated cell types treated with different doses of trastuzumab for five days. Data are presented as the mean ± SD of three technical replicates. *p*-values were determined by RM ANOVA with a post hoc LSD test. n.s., not significant, ** *p* < 0.01, *** *p* < 0.001 versus controls. CON, empty vector; CTTN, CTTN overexpression; shCON, shRNA control; shCTTN, CTTN knockdown. (**E**) Colony formation assays on the indicated cells treated with different doses of trastuzumab for 9 days and maintained in the same medium for another 3 days. (**F**) The effect of CTTN overexpression on the response of HER2+ breast cancer cells to trastuzumab in vivo. Control (CON) or CTTN-overexpressing (CTTN) SKBR3 cells were orthotopically xenografted into mice. The mice were then treated with vehicle (Veh) or 20 mg/kg trastuzumab (Trz). The growth of tumors was monitored twice a week for 5–6 weeks (*n* = 7 mice/group; mean ± SEM). *p*-values were determined with RM ANOVA followed by a post hoc LSD test. * *p* < 0.05 (CON-veh versus CON-Trz), *** *p* < 0.001 (CON-veh versus CTTN-veh or CON-Trz versus CTTN-Trz).

**Figure 3 cancers-15-01168-f003:**
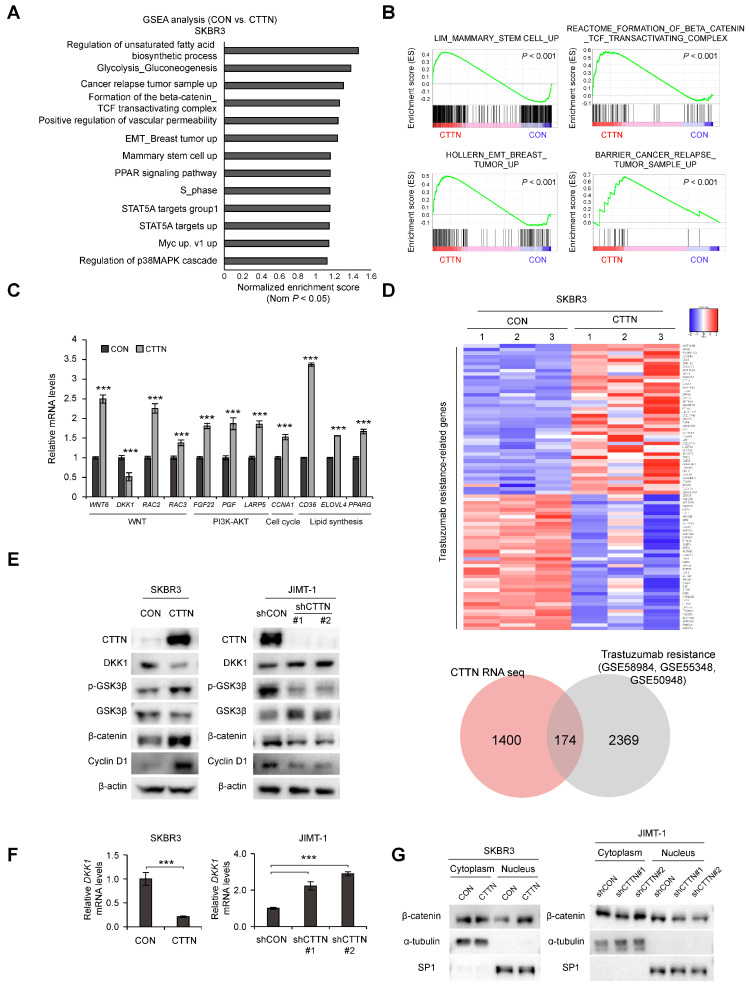
Overexpression of CTTN activates the Wnt signaling pathway by downregulating DKK-1 expression. (**A**,**B**) Analysis of the signaling pathways and gene signatures affected by CTTN overexpression in SKBR3 cells by gene set enrichment analysis (GSEA) of RNA-seq data. (**C**) Validation by qRT-PCR of the CTTN-target genes identified by RNA-seq. Data represent mean ± SD of three technical replicates. *p*-values were determined by two-tailed Student’s *t*-tests. *** *p* < 0.001 versus control. (**D**) Heatmap (top) showing the genes present in both the trastuzumab resistance-associated gene sets (GSE58984 [32], GSE55348 [33], GSE50948 [34]) and the set of genes differentially expressed (≥1.5-fold) in response to CTTN overexpression in SKBR3 cells (as obtained from the RNA seq data). The Venn diagram indicates numbers of overlapping genes. CON, control group; CTTN, CTTN overexpressing group. (**E**) Western blot showing DKK-1, β-catenin and cyclinD1 expression and phosphorylation levels of GSK3-β in SKBR3 and JIMT1 cells. CON, empty vector; CTTN, CTTN overexpression; shCON, shRNA control; shCTTN, CTTN knockdown. (**F**) qRT-PCR analysis of DKK-1 mRNA levels in SKBR3 and JIMT-1 cells. The data are means ± SD of three technical replicates. *p*-values were determined by two-sided Student’s *t*-tests (for SKBR3 cells) or one-way ANOVA with post hoc LSD test (for JIMT-1 cells). *** *p* < 0.001 versus controls (CON or shCON). CON, empty vector; CTTN, CTTN overexpression; shCON, shRNA control; shCTTN, CTTN knockdown. (**G**) Nuclear translocation of β-catenin in the selected stable cell lines. CON, empty vector; CTTN, CTTN overexpression; shCON, shRNA control; shCTTN, CTTN knockdown. The uncropped blots are shown in Appendix A.

**Figure 4 cancers-15-01168-f004:**
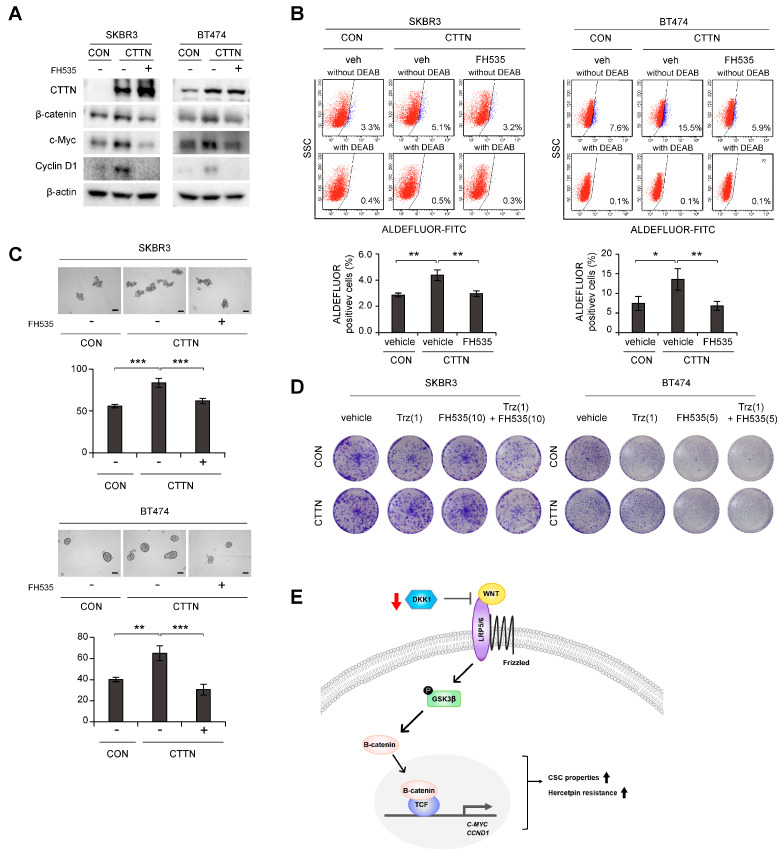
Effect of inhibition of Wnt signaling on CTTN-induced cancer stem cell-like properties and trastuzumab resistance. (**A**) Immunoblots of lysates of the indicated cells after treatment with 10 µM FH535 or DMSO (vehicle) for 24 h. (**B**) Effect of the Wnt/β-catenin/TCF pathway on CTTN-induced CSC activity in HER2+ breast cancer cells. Vehicle (veh) or 10 µM FH535 was added to the cells for 24 h, and the effects on the ALDH+ cell population were assessed by FACS analysis. The data are means ± SD of three technical replicates. *p*-values were determined by one-way ANOVA with post hoc LSD tests. * *p* < 0.05, ** *p* < 0.01. CON, empty vector; CTTN, CTTN overexpression. (**C**) Tumorsphere formation in CTTN-overexpressing SKBR3 and BT474 cells treated with 10 µM FH535. Scale bars, 100 µm. Data means ± SD of three biological replicates. *p*-values were calculated by one-way ANOVA with post hoc LSD tests. ** *p* < 0.01, *** *p* < 0.001. (**D**) Cells were treated with the indicated doses of trastuzumab (µg/mL), FH535 (µM) or Trastuzumab with FH535 for 5 days, maintained in the same medium for 10 days (for SKBR3 cells) or 6 days (for BT474 cells), and colony formation was measured. CON, empty vector; CTTN, CTTN overexpression; Trz, Trastuzumab. (**E**) Proposed model of how CTTN induces Wnt signaling, which in turn stimulates the development of CSC-like properties and trastuzumab resistance. The uncropped blots are shown in Appendix A.

**Table 1 cancers-15-01168-t001:** CTTN enhances tumor-initiating ability in NOD/SCID mice.

Cell Type	Cell Number for Injection	Days						
		25 Days	32 Days	39 Days	47 Days	53 Days	60 Days	62 Days
SKBR3 CON	1000	0/7	0/7	0/7	0/7	0/7	0/7	0/7
	10,000	0/7	1/7	0/7	1/7	2/7	2/7	2/7
	100,000	0/7	2/7	3/7	3/7	5/7	5/7	6/7
TIC frequency			1/221,488	1/204,950	1/151,387	1/65,006	1/65,006	1/45,613
			(1/69,236–1/708,542)	(1/66,312–1/633,440)	(1/55,072–1/416,146)	(1/28,464–1/148,458)	(1/28,464–1/148,458)	(1/20,146–1/103,269)
SKBR3 CTTN	1000	0/7	0/7	0/7	0/7	0/7	0/7	1/7
	10,000	0/7	1/7	4/7	5/7	5/7	5/7	6/7
	100,000	1/7	4/7	5/7	5/7	7/7	7/7	7/7
TIC frequency		1/725,852	1/108,315	1/47,091	1/40,774	1/40,774	1/9538	1/5385
		(1/102,732–1/5,128,513)	(1/43,489–1/269,776)	(1/20,824–1/106,490)	(1/17,876–1/93,000)	(1/17,876–1/93,000)	(1/3924–1/23,184)	(1/2361–1/12,282)
*p*-value			0.343	0.039	0.049	0.433	0.002	<0.001
**Cell Type**	**Cell Number for Injection**	**Days**						
		**21 Days**		**24 Days**		**28 Days**	**31 Days**	
HCC-1954 shCON	1000	0/7		0/7		1/7	4/7	
	10,000	4/7		6/7		7/7	7/7	
	100,000	4/7		6/7		7/7	7/7	
TIC frequency		1/63,227		1/22,866		1/3213	1/1176	
		(1/27,750–1/144,061)		(1/9028–1/57,912)		(1/1339–1/7710)	(1/435–1/3183)	
HCC-1954 shCTTN#1	1000	0/7		0/7		0/7	0/7	
	10,000	0/7		1/7		2/7	2/7	
	100,000	0/7		1/7		2/7	3/7	
TIC frequency		1/725,852		1/359,832		1/164,203	1/119,227	
				(1/86,771–1/1,492,202)		(1/58,039–1/464,560)	(1/46,688–1/304,468)	
*p*-value				0.0015		0.0001	0.0001	

## Data Availability

The RNA-seq data have been deposited in the Gene Expression Omnibus (GEO) database under accession number GSE201563.

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
