# Peer review of "CTTN Overexpression Confers Cancer Stem Cell-like Properties and Trastuzumab Resistance via DKK-1/WNT Signaling in HER2 Positive Breast Cancer"

_cancers, 2023, doi:10.3390/cancers15041168_

Round 1
Reviewer 1 Report
This manuscript is overall well written, despite inconsistent labeling. It addresses a clear and important issue, asks pertinent questions, and uses relatively appropriate methods. However, it has some issues, more specifically:
From Screening western blot images, expression level of CTTN looks similar between SKBR3 cells and HCC1954. This gives me a question; how did authors choose HCC1954 cell line for knockdown experiments?
We do not see JIMT-1 cell line in screening western blot experiment, I suggest authors to include it and show in screening western blot. Although it’s a supplementary data, whole manuscript is relying on this expression data.
Figure 3A: Normalized Enrichment Score should be on x-axis. Also, I would suggest authors to show if not all at least some top upregulated and down regulated genes and gene sets from RNA-seq data in supplementary file.
Although, authors have presented all original blots. Some blots are dirty, making it difficult to interpret results. I suggest quantifying western blots with imageJ and adding this information to supplementary data.
Minor issue: Table1: Is overlapping with line numbers, making it difficult to read the table.
Reviewer 2 Report
Moon et al investigated the role of CTTN overexpression in HER2-positive breast cancer. They revealed prognostic relevance of CTNN expression in tissue and investigated the cancer stem cell properties as well as possible mechanisms of drug resistance towards tratsuzumab. This is a solid piece of work and many experiments were performed. However, the manuscript needs to be thoroughly restructured. It should become clear which parts derived from cell-cultures, xenografts, patients and publicly available datasets. The results should be structured accordingly. It would be helpful for the reader if the authors could give some background on their comprehensive approach.
Comments:
Simple summary: Should be re-written in a more scientific way. It does not become clear which findings were obtained from the literature or from the results of the study. Does the combinational treatment refer to the patients’ cohort or in vivo experiments? CTTN should be introduced.
Abstract: Line 20: How many patients were enrolled in the “multiple breast cancer cohorts”? Does overexpression refer to IHC or mRNA? What is KM plotter? Kaplan Meier survival curve? How many xenografts? What was used in vivo – tissue or cells? What did the colony formation assay show? Results should match the methods. Line 30: Which assay was leading to this result?
Methods: Many cell lines were used. It would be interesting to mention a few aspects on the selection and type of cells.
2.6, 2.7., 2.8., 2.9. What did the authors investigate using these assays? In addition, which drug concentrations were used?
2.11. What is B27?
2.13. What samples were used? Cell-culture or tumors from xenografts or patients’ tumor tissue?
Statistical Analysis: Pearson correlation was used but not mentioned.
It should be clarified in the abstract which data were received from public databases and which data was generated experimentally.
P.5 Line 237: How many patients had HER2-enriched breast cancer of the used sets: METABRIC (n=247) and TCGA (n=120) breast cancer datasets? In Figure 1 there are 243 patients with HER2+ derived from Metabric. What is the difference between the two databases?
Figure 1: It should be explained which cohorts were picked and how sub-cohorts were defined. Which cut-off values were chosen to distinguish between high and low CTTN expression? The Kaplan-Meier analyses should be re-considered as 300 months survival time does not make sense. Could the survival data from Metabric be confirmed with the patients from the TCGA? Here, a little background on the entire approach would help to understand the results. Figure 1D: Which expression does this refer to? I do not understand Figure 1E. If only 236 patients were observed, how can there be 649 low? Figure 1 F needs to be improved. What is HNU? How many patients? 400x magnification?? Scale bar missing.
3.2. FACS not mentioned yet. This is a very interesting and well-written paragraph. Should be placed between Figure 1 and Figure 2, though.
I cannot find Table 1 in the main text. What exactly is the TIC frequency telling me? What was the calculation based on?
In general: Were the Trastuzumab concentrations close to clinically applied doses? Did FH535 cause any toxicity? What would be a possible way to deliver it into patients?
Reviewer 3 Report
Dear Authors,
Congratulation on your article. It includes a novel topic, interesting scientific data, wonderful graphic.
I don t have any remarks on the context of the article.
Some details can be improved:
the legends of the figures are too long
the Conclusion section is too short. I encourage you maybe to enlarge even the discussion part.
English can be a little improved.
most of the references are old
Kind regards
